# Fucoxanthin Ameliorates Oxidative Stress and Airway Inflammation in Tracheal Epithelial Cells and Asthmatic Mice

**DOI:** 10.3390/cells10061311

**Published:** 2021-05-25

**Authors:** Shu-Ju Wu, Chian-Jiun Liou, Ya-Ling Chen, Shu-Chen Cheng, Wen-Chung Huang

**Affiliations:** 1Department of Nutrition and Health Sciences, Research Center for Chinese Herbal Medicine, Chang Gung University of Science and Technology, Taoyuan City 33303, Taiwan; sjwu@mail.cgust.edu.tw; 2Aesthetic Medical Center, Department of Dermatology, Chang Gung Memorial Hospital, Linkou, Taoyuan City 33303, Taiwan; 3Department of Nursing, Division of Basic Medical Sciences, Research Center for Chinese Herbal Medicine, Chang Gung University of Science and Technology, Taoyuan City 33303, Taiwan; ccliu@mail.cgust.edu.tw; 4Division of Allergy, Asthma, and Rheumatology, Department of Pediatrics, Chang Gung Memorial Hospital, Linkou, Guishan Dist., Taoyuan City 33303, Taiwan; 5School of Nutrition and Health Sciences, Taipei Medical University, Taipei City 11031, Taiwan; ylchen01@tmu.edu.tw; 6Department of Traditional Chinese Medicine, Chang Gung Memorial Hospital, Taoyuan City 33378, Taiwan; 7Graduate Institute of Health Industry Technology, Research Center for Food and Cosmetic Safety, Research Center for Chinese Herbal Medicine, College of Human Ecology, Chang Gung University of Science and Technology, Taoyuan City 33303, Taiwan

**Keywords:** airway hyperresponsiveness, asthma, fucoxanthin, oxidative stress, tracheal epithelial cells

## Abstract

Fucoxanthin is isolated from brown algae and was previously reported to have multiple pharmacological effects, including anti-tumor and anti-obesity effects in mice. Fucoxanthin also decreases the levels of inflammatory cytokines in the bronchoalveolar lavage fluid (BALF) of asthmatic mice. The purpose of the present study was to investigate the effects of fucoxanthin on the oxidative and inflammatory responses in inflammatory human tracheal epithelial BEAS-2B cells and attenuated airway hyperresponsiveness (AHR), airway inflammation, and oxidative stress in asthmatic mice. Fucoxanthin significantly decreased monocyte cell adherence to BEAS-2B cells. In addition, fucoxanthin inhibited the production of pro-inflammatory cytokines, eotaxin, and reactive oxygen species in BEAS-2B cells. Ovalbumin (OVA)-sensitized mice were treated by intraperitoneal injections of fucoxanthin (10 mg/kg or 30 mg/kg), which significantly alleviated AHR, goblet cell hyperplasia and eosinophil infiltration in the lungs, and decreased Th2 cytokine production in the BALF. Furthermore, fucoxanthin significantly increased glutathione and superoxide dismutase levels and reduced malondialdehyde (MDA) levels in the lungs of asthmatic mice. These data demonstrate that fucoxanthin attenuates inflammation and oxidative stress in inflammatory tracheal epithelial cells and improves the pathological changes related to asthma in mice. Thus, fucoxanthin has therapeutic potential for improving asthma.

## 1. Introduction

Bronchial asthma is one of the most common respiratory allergies and inflammatory diseases globally. Severe air pollution produces more suspended particles and chemical irritants and increases the prevalence of asthma and respiratory disease in developing and developed countries [1]. The characteristics of sudden asthma include chest tightness, difficulty breathing, shortness of breath, dry cough, and paroxysmal wheezing [2]. Many studies have pointed out that the airways of patients with chronic asthma are excessively sensitive to allergens, chemical irritants, and particle pollution. Other studies have confirmed that mites, cigarette smoke, pollen, cold air, and ozone may stimulate sudden allergic reactions in the airways and asthma attacks [3]. Continuous inflammation of the respiratory tract is an important factor that worsens the pathological symptoms of asthma. In patients with asthma, allergic stimulants also induce hyperplasia and sensitivity of the tracheal epithelial cells. These tracheal epithelial cells increase mucus secretion, leading to airway obstruction, breathing difficulties, and even suffocation [4].

Recent studies have shown that activated T cells and inflamed macrophages in the lungs of asthmatic patients release high amounts of IL-4 and TNF-α [5]. These cytokines lead to the inflammation and oxidation of the tracheal epithelial cells and the secretion of more inflammatory cytokines, chemokines, and eotaxins. Eotaxins not only attract more eosinophils to infiltrate the lungs, but inflammatory cytokines and chemokines also cause more severe airway inflammation, oxidative cell damage, and airway remodeling, leading to worsening lung function and aggravated breathing difficulties in patients with asthma [6]. Therefore, inflamed tracheal epithelial cells play an important role in the pathological features of patients with acute asthma and severe persistent asthma.

Many studies have confirmed that the deterioration of allergic asthma is closely related to immune system disorders or imbalanced immune cell activation. Massive proliferation and activation of Th2 cells and ILC2 are important factors in the induction of immune system disorders in asthma patients [7]. Excessive activation of Th2 cells results in the release of a large amount of cytokines IL-4, IL-5, and IL-13, causing eosinophil infiltration into the lungs, airway hyperresponsiveness (AHR), and the induction of mast cell and eosinophil release of inflammatory, oxidative, and allergic molecules to damage the epithelial cells in airways [5]. Th2 cell-associated cytokines can also induce goblet cell proliferation in the trachea, stimulating the secretion of large amounts of mucus to block the airways [8]. In sudden asthma attacks, the airway epithelial cells secrete more inflammatory chemokines to attract more immune cells to infiltrate the lungs, including neutrophils, macrophages, and eosinophils. These activated immune cells and airway epithelial cells may release excess reactive oxygen species (ROS), exacerbating the inflammation and oxidative damage to the lungs of asthmatic patients and causing damage to the lung cells and tissues [9]. In the lungs of asthma patients, the synergistic effect of excessive ROS and IL-13 leads to more serious AHR and airway remodeling, increase the thickness of airway smooth muscle, and reduce the gas exchange rate of the respiratory system [7]. Therefore, reducing the activity of Th2 cells in the lungs and the expression of ROS and inflammation-related signaling molecules in the lung and airway epithelial cells is a strategy to improve the development of asthma.

Clinically, drugs commonly used to treat or prevent asthma include bronchodilators as well as anti-inflammatory and anti-allergy drugs. Inhaled steroids and oral steroids are commonly used to treat or prevent asthma, but some patients have side effects on steroids [10]. The use of steroid drugs is limited and invalid for treating neutrophilic asthma or severe asthma patients [11]. Therefore, the development of new effective drugs for treating asthma is expected by many researchers and clinicians.

Some traditional Chinese herbs and pure plant compounds have been found to improve asthma [12,13,14]. However, the relationship between the pure compounds or extracts isolated from algae and the improvement of asthma symptoms has been studied infrequently. Fucoxanthin is a carotenoid pigment and is abundant in brown seaweeds (Phaeophyceae) and marine diatoms (Bacillariophyta) [15]. Fucoxanthin has multiple biological functions and potential health benefits in animal models and cell experiments [16,17,18]. Fucoxanthin has been demonstrated to inhibit inflammatory mediators and pro-inflammatory cytokines in LPS-stimulated macrophages, and it could decrease lung fibrosis in bleomycin-induced mice [19,20]. Furthermore, fucoxanthin could decrease ROS expression by promoting neuroprotective effects by regulating the Nrf2-autophagy pathways in a mouse model of traumatic brain injury [21]. Recently, fucoxanthin was found to decrease inflammatory cytokine expression in the bronchoalveolar lavage fluid (BALF) of asthmatic mice and reduce the levels of total IgE, histamine, and malondialdehyde (MDA) in the serum of mice with allergic rhinitis [22,23]. However, whether fucoxanthin reduces oxidative stress in the lungs, goblet cell hyperplasia, and mucus hypersecretion in the trachea is still unclear. In the current study, we examined whether fucoxanthin attenuates the oxidative responses and inflammatory cytokine expression in human tracheal epithelial cells. We also investigated whether fucoxanthin ameliorated the molecular mechanisms underlying oxidative stress in the lungs, airway inflammation, and pathological lesions in the lungs of asthmatic mice.

## 2. Materials and Methods

### 2.1. Materials

Fucoxanthin (the purity was ≥95% by HPLC) was purchased from Sigma Aldrich (St. Louis, MO, USA). For cell experiments, the concentration of the stock solution was 100 mM, with DMSO as the solvent. DMSO was ≤0.1% of the culture medium as described previously [24]. For asthmatic animal experiments, fucoxanthin was dissolved in DMSO, and the working solution was formulated as 10 mg/50 μL and 30 mg/50 μL.

### 2.2. Cell Viability Assay

Cell viability was assayed using cell counting kit-8 (CCK-8, Sigma) as described previously [25]. Cells (10^4^/well) were seeded in 96-well culture plates and treated with various concentrations of fucoxanthin for 24 h. Cells treated with the CCK-8 solution were used to determine cell viability using a microplate reader (Multiskan FC, Thermo, Waltham, MA, USA).

### 2.3. BEAS-2B Cell Culture and Fucoxanthin Treatment

Human bronchial epithelial cells (BEAS-2B; American Type Culture Collection, Manassas, VA, USA) were seeded into 24-well culture plates in DMEM/F12 medium supplemented with 10% FBS and 100 U/mL penicillin and streptomycin. BEAS-2B cells were treated with various concentrations of fucoxanthin (0–30 μM) for 1 h. Subsequently, the cells were induced with 10 ng/mL TNF-α and 10 ng/mL IL-4 for 24 h. The supernatants were collected, and the chemokine or cytokine levels were detected using specific ELISA kits.

### 2.4. Cell-Cell Adhesion Assay

BEAS-2B cells were treated with fucoxanthin and incubated with 10 ng/mL TNF-α/IL-4 for 24 h. Human monocytic THP-1 cells were purchased from the Bioresource Collection and Research Center (BCRC, Taiwan) and cultured in RPMI 1640 medium. The THP-1 cells were treated with calcein-AM solution (Sigma) for 30 min as described previously [26]. We co-cultured the THP-1 and BEAS-2B cells and detected the adherent THP-1 cells using fluorescence microscopy (Olympus, Tokyo, Japan). 

### 2.5. ROS Production in BEAS-2B Cells

BEAS-2B cells were treated with fucoxanthin for 1 h and incubated with TNF-α/IL-4 for 24 h. Next, the BEAS-2B cells were stimulated with 2′,7′-dichlorofluorescin diacetate (DCFH-DA) for 30 min, and the intracellular ROS was observed using fluorescence microscopy (Olympus) as described previously [26]. Moreover, the cells were lysed, and the ROS production was detected, both using a multi-mode microplate reader (BioTek synergy HT, Winooski, VT, USA). 

### 2.6. Animal Experiments

Six-week-old female BALB/c mice were purchased from the National Laboratory Animal Center (Taipei, Taiwan). All mice were housed in temperature-controlled animal housing under a 12-h light-dark cycle and were raised with food and water available ad libitum. The animal experiments were approved and carried out in accordance with the guidelines from the Laboratory Animal Care Committee of Chang Gung University of Science and Technology (IACUC approval number: 2019-003). Mice were randomly divided into the five following experimental groups (*n* = 10 each): normal control group (N) mice were sensitized with normal saline and treated with DMSO by intraperitoneal injection; OVA control group (OVA) mice were sensitized with OVA and treated with DMSO by intraperitoneal injection; prednisolone positive control group (P) mice were sensitized with OVA and treated with 5 mg/kg prednisolone by intraperitoneal injection; or fucoxanthin experiment groups, in which OVA-sensitized mice were treated with 10 mg/kg or 30 mg/kg fucoxanthin (Fu10 and Fu30 groups, respectively) by intraperitoneal injection.

### 2.7. Mouse Sensitization and Administration of Fucoxanthin

Mice were treated with or without the sensitized solution containing 0.8 mg aluminum hydroxide (Thermo, Rockford, IL, USA) and 50 μg ovalbumin (OVA; Sigma) in 200 μL normal saline by intraperitoneal injections on days 1–3 and 14. Subsequently, mice were challenged with 2% OVA via an inhaled atomized vapor for 30 min on days 14, 17, 20, 23, and 27 using an ultrasonic nebulizer. The mice were treated with fucoxanthin, prednisolone, or DMSO solution by intraperitoneal injection 1 h before the challenge of OVA or methacholine (Sigma) inhalation (on day 28). AHR was detected on day 28, and the mice were sacrificed to evaluate oxidative stress, inflammatory response, asthma pathology, and immune regulation on day 29.

### 2.8. Airway Hyperresponsiveness

AHR was assessed to demonstrate airway function as described previously [27]. Mice were put in a single chamber and allowed to inhale 0 to 40 mg/mL aerosolized methacholine to detect the enhanced pause (Penh) using whole-body plethysmography (Buxco Electronics, Troy, NY, USA).

### 2.9. Histological Analysis of Lung Tissue 

Lung tissues were removed and fixed with 10% formalin before being embedded in paraffin and cut into 6-μm sections. A section of lung biopsy was treated with Masson’s trichrome stain to detect collagen expression. The lung section was also stained using hematoxylin and eosin (HE) solution to examine the eosinophil infiltration of the lungs using a 5-point scoring system and stained with periodic acid-Schiff (PAS) solution (Sigma) to observe the goblet cell hyperplasia of the trachea as described previously [27].

### 2.10. Serum Collection and Splenocyte Culture

Mice were anesthetized with isoflurane and blood was collected from the orbital vascular plexus. The blood was centrifuged at 6000 rpm for 5 min; the serum was then collected and stored at −80 °C. The serum would detect OVA-specific antibody expression by ELISA as described previously [28]. In addition, 5 × 10^6^ splenocytes/mL were incubated with 100 μg/mL OVA for 5 continuous days, and cytokine levels were detected using a specific ELISA kit as described previously [29].

### 2.11. Bronchoalveolar Lavage Fluid and Cell Counting 

The BALF was collected as described previously [30]. Mice were anesthetized and sacrificed using an indwelling needle to intubate the trachea to wash the lungs and airways. The lavage fluid was centrifuged, and the supernatant was collected to detect cytokine and chemokine levels. We used Giemsa stain solution (Sigma) to identify the morphology of the different immune cells.

### 2.12. RNA Isolation and Quantitative Real-Time PCR 

Lung tissues were homogenized, and RNA were extracted, both using TRI reagent (Sigma). Using the cDNA synthesis kit (Bio-Rad, San Francisco, CA, USA), we synthesized the cDNA and investigated the specific gene expression using SYBR Green in the quantitative real-time PCR procedure using a spectrofluorometric thermal cycler (iCycler; Bio-Rad).

### 2.13. ELISA

Supernatant from BEAS-2B cell culture medium and BALF was used to detect CCL5, CCL11, CCL24, MCP-1, TNF-α, IL-4, IL-5, IL-6, IL-8, and IL-13 levels using specific ELISA kits (R&D Systems, Minneapolis, MN, USA) as described previously [28]. Serum OVA-specific IgG1, IgG2a, and IgE were assayed by specific ELISA kits (BD Biosciences, San Diego, CA, USA). OVA-IgG1 and OVA-IgG2a standard curves were obtained using serum from OVA-sensitized mice; the serum was diluted 5-fold to observe OVA-IgE by measuring absorbance at 450 nm.

### 2.14. Immunohistochemical Staining

Paraffin-embedded sections of lung tissues were incubated with a specific COX-2 antibody (1:100; ab15191, Abcam, Cambridge, UK) overnight, followed by a secondary antibody. The slides were treated with DAB substrate solution to detect COX-2 expression as described previously [31].

### 2.15. MDA Activity 

MDA activity in the lungs was detected using a lipid peroxidation assay kit (Sigma) according to the manufacturer’s instructions. The lung tissues were homogenized using a homogenizer (FastPrep-24, MP Biomedicals, Santa Ana, CA, USA) and treated with perchloric acid (150 µL, 2N) for protein precipitation. The samples were centrifuged to collect the supernatant, and MDA activity was detected using a multi-mode microplate reader (BioTek Synergy HT). 

### 2.16. Glutathione (GSH), Superoxide Dismutase (SOD), and Catalase (CAT) Assay 

We used a glutathione assay kit, a superoxide dismutase assay kit, and a catalase assay kit (Sigma) to detect the levels of GSH, SOD, and CAT in the lung tissues according to the manufacturer’s instructions. 

### 2.17. Statistical Analysis

Statistical analysis was performed using one-way analysis of variance (ANOVA) followed by a Kruskal–Wallis test. All data are expressed as the mean ± standard error of the mean (SEM), and at least three independent experiments were analyzed. A *p* value < 0.05 was considered significant.

## 3. Results

### 3.1. Fucoxanthin Reduced Inflammatory Mediators and Cell Adhesion in BEAS-2B Cells 

The cytotoxicity of fucoxanthin in the BEAS-2B cells was determined using the CCK8 assay. Fucoxanthin did not demonstrate significant cytotoxic effects at a concentration ≤50 μM, and subsequent experiments used fucoxanthin at 0–30 μM (Figure 1A). In the BEAS-2B cells, fucoxanthin significantly decreased the levels of CCL5, CCL11, CCL24, IL-6, IL-8, and MCP-1 compared to the TNF-α/IL-4-stimulated BEAS-2B cells (Figure 1B–G). Fucoxanthin also reduced THP-1 cell adherence to the TNF-α/IL-4-activated BEAS-2B cells (Figure 2A,B).

### 3.2. Effect of Fucoxanthin on ROS Production 

Fluorescence microscopy showed that fucoxanthin reduced intracellular ROS production compared to TNF-α/IL-4-stimulated BEAS-2B cells (Figure 3A,B). Furthermore, in the BEAS-2B cells treated with DCFH-DA, fucoxanthin significantly attenuated ROS levels in the TNF-α/IL-4-stimulated BEAS-2B cells (Figure 3C). 

### 3.3. Fucoxanthin Attenuated AHR in Asthmatic Mice

The process of sensitization and asthma induction is shown in Figure 4A. Penh values can be used as indicators of AHR. Mice inhaled gradually increasing doses of methacholine (0–40 mg/mL) to assess whether fucoxanthin could recover airway function in an OVA-induced allergic asthma murine model. The Penh values significantly increased when OVA-sensitized mice inhaled gradually increasing doses of methacholine compared to normal mice (Figure 4B). Inhalation of high-dose methacholine (40 mg/mL), fucoxanthin treatment, or prednisolone treatment significantly reduced the Penh values compared to the OVA group (7.99 ± 0.47 vs. P: 4.10 ± 0.18, *p* < 0.01; Fu10: 6.08 ± 0.46, *p* < 0.05; and Fu30: 5.09 ± 0.35, *p* < 0.01). Therefore, the experimental results confirmed that fucoxanthin can significantly decrease AHR in asthmatic mice.

### 3.4. Fucoxanthin Inhibited Eosinophils in the BALF

Inflammatory cells in the BALF stained with Giemsa solution in the OVA group of asthmatic mice had higher numbers of eosinophils than normal mice. The OVA-sensitized asthmatic mice treated with fucoxanthin had significantly reduced numbers of eosinophils compared to the OVA control group. We also found that the total number of cells in asthmatic mice was significantly reduced after treatment with fucoxanthin (Fu10: 1.13 × 10^6^ ± 8.75 × 10^4^, *p* < 0.01; Fu30: 1.02 × 10^6^ ± 4.97 × 10^4^, *p* < 0.01) or prednisolone (7.72 × 10^5^ ± 4.98 × 10^4^, *p* < 0.01) compared to the OVA control group (1.49 × 10^6^ ± 6.21 × 10^4^; Figure 4C).

### 3.5. Fucoxanthin Modulated Chemokine and Cytokine Expression in the BALF and Lung Tissue

In the BALF, fucoxanthin significantly inhibited the levels of TNF-α, IL-4, IL-5, IL-6, IL-13, CCL11, and CCL24 compared to the OVA-sensitized mice (Figure 5). Fucoxanthin also resulted in significantly higher IFN-γ expression in the BALF than in the OVA group (Figure 5D). In the lung tissues, fucoxanthin significantly suppressed MUC5AC, TNF-α, IL-6, IL-4, IL-5, IL-13, CCL11, and CCL-24 expression and significantly increased IFN-γ expression compared to the OVA-sensitized asthmatic mice (Figure 6). 

### 3.6. Fucoxanthin Reduced Goblet Cell Hyperplasia and Eosinophil Infiltration in Lung Tissue

HE staining demonstrated that OVA-sensitized mice treated with fucoxanthin or prednisolone had reduced eosinophil infiltration of the lungs compared to the OVA control group (Figure 7A). Thus, fucoxanthin and prednisolone significantly decreased the inflammatory pathology score in the OVA-sensitized asthmatic mice (Figure 7B). Goblet cells were detected in the trachea by PAS staining, and fucoxanthin or prednisolone treatment of the OVA-sensitized mice significantly reduced goblet cell hyperplasia compared to the OVA control group (Figure 7C,D).

### 3.7. Fucoxanthin Reduced Collagen and COX-2 Expression in the Lungs

Lung collagen accumulation was indicated by Masson’s trichrome stain. Fucoxanthin decreased collagen expression in the lung tissues from asthmatic mice (Figure 7E,F). Moreover, fucoxanthin significantly decreased the gene expression of COX-2 in the lungs of OVA-sensitized mice (Figure 8A). Subsequently, immunohistochemical staining demonstrated that fucoxanthin could decrease COX-2 expression in the lung tissue compared to the OVA control group (Figure 8B).

### 3.8. Fucoxanthin Modulated Antioxidant Enzyme Levels in the Lungs

Fucoxanthin significantly increased GSH, SOD, and CAT expression and decreased MDA activity in the lung tissues compared to the OVA-sensitized mice (Figure 9A–D). 

### 3.9. Fucoxanthin Modulated Serum Antibody and Splenocyte Cytokine Levels

In the serum from OVA-sensitized mice, fucoxanthin significantly decreased OVA-IgG1 and OVA-IgE and increased OVA-IgG2a levels (Figure 10A–C). In the supernatant from splenocyte cultures, fucoxanthin also clearly reduced IL-4, IL-5, and IL-13 expression and promoted IFN-γ expression compared to OVA-sensitized mice (Figure 10D–G).

## 4. Discussion

Brown seaweeds contain a variety of biologically active compounds, including polyphenols, omega-3 polyunsaturated fatty acids, fucosterol, fucoidan, and fucoxanthin [32]. Fucoxanthin is a carotenoid, an orange-red pigment found in the chloroplasts [16]. Recent studies have shown that fucoxanthin has anti-inflammatory, anti-oxidant, and anti-tumor effects in cell and animal experiments [15,18,20,33]. Fucoxanthin also inhibits the expression of inflammatory cytokines and inflammatory signaling molecules in LPS-induced macrophages, and can decrease oxygen-glucose deprivation and re-oxygenation-induced ROS expression in neurons [33]. Fucoxanthin could protect cerebral ischemic/reperfusion injury rats from nerve inflammation and oxidative stress by promoting the Nrf2/HO-1 signaling pathway [21]. Previous studies have shown that 50 mg/kg fucoxanthin can reduce the inflammatory cytokine levels in the BALF of mice [22]. Furthermore, in the mouse allergic rhinitis model induced by OVA, fucoxanthin treatment reduces the MDA levels in the nasal mucosa and serum, and reduces eosinophil infiltration in the nasal cavity [23]. However, whether fucoxanthin can improve lung inflammation and antioxidant effects in the tracheal epithelial cells and asthmatic mice is unclear. In addition, in the current study, the mice were treated with different doses of fucoxanthin for 7 consecutive days for animal toxicity testing, which demonstrated that daily administration of 100 mg/kg, 50 mg/kg, and 30 mg/kg fucoxanthin caused 37.5%, 12.5%, and 0% mortality, respectively (data not shown). We also found that the mice treated with 100 mg/kg and 50 mg/kg fucoxanthin by intraperitoneal injection had significantly lower activity and appetite than the normal mice and the mice treated with 30 mg/kg fucoxanthin (data not shown). Therefore, the animal experiments used 10 mg/kg or 30 mg/kg fucoxanthin to investigate whether fucoxanthin can improve the asthma pathology in OVA-sensitized asthmatic mice. We also evaluated the molecular mechanism by which fucoxanthin reduces airway inflammation and oxidative stress in asthmatic mice and analyzed the effect of fucoxanthin on inflammation and oxidative stress in inflamed tracheal epithelial cells.

Our experiments used TNF-α/IL-4 to stimulate human tracheal epithelial cells, simulating lung inflammation immune cells that release TNF-α and IL-4 to induce tracheal epithelial cell activation. Under the protection of fucoxanthin treatment, we found that TNF-α/IL-4-stimulated BEAS-2B cells had reduced IL-6 secretion to inhibit the inflammatory response in the airways. Previous experiments in asthmatic animals have found that ROS can stimulate the activation of the airway and tracheal epithelial cells, and the release of more chemokines to attract inflammatory immune cell infiltration into the lungs for the release of more inflammatory mediators, causing lung cell damage and apoptosis [34,35]. In the current study, under the protection of fucoxanthin treatment, we found that TNF-α/IL-4-stimulation in BEAS-2B cells reduced the release of IL-8, MCP-1, and CCL5, which would contribute to a reduction in macrophage and neutrophil infiltration of the lungs as well as inflammation and oxidative damage in the lung tissue. Therefore, we think that fucoxanthin could inhibit the inflammation of tracheal epithelial cells to release more ROS and chemokines, reducing lung cell damage. 

Asthma attacks are mainly induced by allergens or airway irritants to cause acute allergies and inflammatory reactions in the airways. However, chronic asthma is often caused by long-term allergic stimulation of the airways, causing inflammation and oxidative damage, and gradually weakening the physiological function of the respiratory system [36]. We found that fucoxanthin can reduce ROS levels in inflamed BEAS-2B cells, indicating that fucoxanthin can reduce oxidative damage to tracheal epithelial cells. In addition, fucoxanthin inhibited the expression of MCP-1, CCL5, and IL-8, which would suppress the migration of neutrophils or macrophages into the lungs. Therefore, we think that fucoxanthin can inhibit the activation of inflammatory tracheal epithelial cells and reduce ROS release and chemokine secretion in tracheal epithelial cells.

SOD, CAT, and GSH are common antioxidant enzymes that can reduce lung cell damage and fibrosis in the lungs of patients with COPD or asthma [37]. Lipid peroxidation is a danger signal of cell damage and would produce abundant MDA as a marker of oxidative stress in cells and tissues [38,39]. In the current study, fucoxanthin promoted SOD, CAT, and GSH expression and decreased MDA expression in the lungs of asthmatic mice (Figure 9). Thus, fucoxanthin could regulate and improve lung peroxidation in asthmatic mice. Previous studies found that fucoxanthin can enhance the Nrf2/HO-1 pathway to improve GSH and SOD expression and reduce brain damage in rat models of traumatic brain injury [21]. Therefore, fucoxanthin has a protective ability against oxidative stress for maintaining lung function in asthmatic mice.

AHR can detect the airflow and frequency of breathing, and has clinical value as an important indicator to evaluate lung function [40]. Previous studies have found that inflammation and oxidative stress stimulate excessive AHR and exacerbate the deterioration of lung function [41]. Allergens or respiratory irritants can induce excessive tracheal contraction and mucus hypersecretion in asthmatic patients, leading to airway obstruction and dyspnea during asthma attacks. Therefore, patients need to breathe more quickly to increase the rate of ventilation, resulting in an increased AHR value in patients with asthma [42]. We used whole body plethysmography to detect AHR in mice and demonstrated that fucoxanthin reduced the AHR and the breathing difficulties caused by rapid breathing in asthmatic mice. Many previous studies have pointed out that the lungs of asthmatic patients have increased IL-13 expression, and activated Th2 cells increase IL-13 secretion to promote AHR [43]. Our experiments showed that the lung tissues, BALF, and spleen cell culture medium for asthmatic mice had significantly reduced IL-13 levels and reduced AHR values after treatment with fucoxanthin (Figure 4, Figure 5, Figure 6 and Figure 10). Therefore, our results confirmed that fucoxanthin mainly inhibited IL-13 production to reduce AHR in asthmatic mice.

Asthma patients have a lot of eosinophilic infiltration in the lungs and respiratory tract. Activated Th2 cells in the lungs of asthmatic patients release excess IL-5, which induces the differentiation of bone marrow cells to produce mature eosinophils [44]. In addition, tracheal epithelial cells from asthmatic patients release eotaxins (CCL11 and CCL24) to attract mature eosinophil migration into the lungs [45]. The activated eosinophils release a large amount of basic proteins, stimulate mast cell activation and degranulation, and release allergic mediators to cause allergic reactions in the lungs of asthmatic patients [5]. In addition, IL-4 could activate B-cell differentiation and secrete IgE to bind the FcεR1 of mast cells. After exposure to allergens, the mast cells are activated to release leukotrienes and histamine, causing respiratory sensitization and inflammation [46]. Therefore, fucoxanthin reduces the production of IL-4 and IgE by inhibiting Th2 cell activity, decreasing mast cell activation, and improving the pathological changes in bronchial asthma. Previous researchers have found that fucoxanthin can suppress the levels of histamine and total IgE in serum from allergic rhinitis mice [23]. In our asthmatic mouse model, the BALF and lungs expressed higher IL-5 and eotaxin levels than normal mice (Figure 5 and Figure 6). Fucoxanthin has the ability to reduce IL-5 production to suppress eosinophil infiltration in the lungs of asthmatic mice. In addition, fucoxanthin inhibits CCL11 and CCL24 levels in tracheal epithelial cells and lung tissue, which reduces eosinophil migration into the lungs. Therefore, fucoxanthin could reduce the infiltration of eosinophils into the lungs and BALF of asthmatic mice. 

THP-1 cells simulated the adherence of immune cells in the inflamed tracheal epithelial cell model. Fucoxanthin reduced the adherence of THP-1 cells to tracheal epithelial cells (Figure 2). We think that fucoxanthin could inhibit IL-5 secretion in the lung and BALF to reduce the inflammatory response of tracheal epithelial cells to release eotaxins. 

The airway epithelial cells secreted appropriate glycoprotein mucus that could adhere to air pollution particles and microorganisms, reducing damage in the lungs and respiratory tract [9]. However, the airway remodeling in asthmatic patients leads to the thickening of the tracheal smooth muscle and the narrowing of the airways [4]. Induced by allergens or chemical stimuli, tracheal epithelial cells increase and differentiate into sensitive goblet cells. These goblet cells secrete excessive mucus and block the airways [47]. Our experiments used PAS staining to observe the goblet cell hyperplasia in asthmatic mice and demonstrated that fucoxanthin significantly reduces tracheal goblet cell hyperplasia and decreases Muc5Ac gene expression in the lungs of asthmatic mice (Figure 6 and Figure 7). Therefore, fucoxanthin can reduce the symptoms of dyspnea caused by excessive secretion of respiratory mucus in asthmatic mice. 

Previous studies found that IL-4 and IL-13 released by activated Th2 cells increase goblet cell proliferation in the tracheas of asthmatic mice [48]. However, fucoxanthin has the ability to reduce the expression of IL-4 and IL-13 in spleen cell culture medium, BALF, and lung tissue from asthmatic mice. Therefore, we think that fucoxanthin could reduce the secretion of IL-4 and IL-13 by Th2 cells to reduce the proliferation of goblet cells in the trachea and reduce the breathing difficulties caused by excessive mucus secretion.

Continued inflammation and oxidative damage will increase collagen deposition in the lungs and cause severe pulmonary fibrosis, worsening the respiratory physiology of the lungs [49]. The BALF and lungs of asthmatic patients have increased expression of TGF-β and lung tissue fibrosis [50]. Some studies have pointed out that IL-13 would also drive the expression of TGF-β, exacerbating lung fibrosis [5,51]. Anti-IL-13 (tralokinumab) can improve lung fibrosis in asthmatic patients [52]. In the current study, fucoxanthin significantly inhibited collagen deposition in the lungs of asthmatic mice (Figure 7). Therefore, we speculate that fucoxanthin improves pulmonary fibrosis and AHR in asthmatic mice by reducing the activity of IL-13.

## 5. Conclusions

Collectively, we confirmed that fucoxanthin significantly suppresses AHR, eosinophil infiltration, and mucus secretion by suppressing airway inflammation, oxidative stress, and Th2 cytokine expression in asthmatic mice. Fucoxanthin also decreased ROS, pro-inflammatory cytokine, and eotaxin levels in BEAS-2B cells, and suppressed monocyte cell adherence to inflammatory BEAS-2B cells. Therefore, this study demonstrates that fucoxanthin has excellent potential for ameliorating or regulating inflammation and oxidative stress in asthma.

## Figures and Tables

**Figure 1 cells-10-01311-f001:**
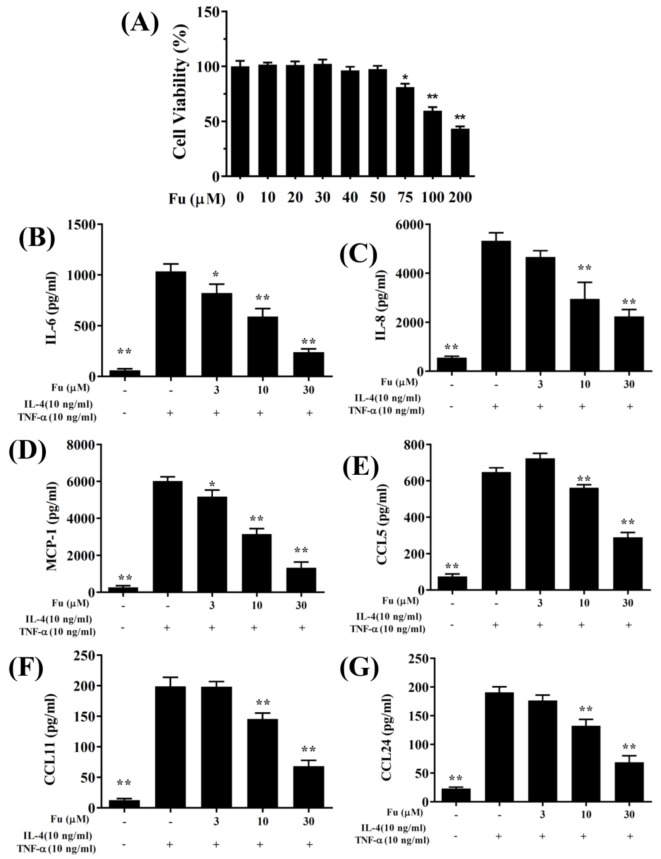
The effects of fucoxanthin (Fu) on cytokine and chemokine production in BEAS-2B cells. (**A**) Cell viability with increasing concentrations of Fu. (**B**) ELISA results for IL-6, (**C**) IL-8, (**D**) MCP-1, (**E**) CCL5, (**F**) CCL11, and (**G**) CCL24 levels in BEAS-2B cells treated with TNF-α/IL-4 and/or Fu. Three independent experiments were analyzed. Data are presented as the mean ± SEM. * *p* < 0.05, ** *p* < 0.01 compared to BEAS-2B cells stimulated with TNF-α and IL-4.

**Figure 2 cells-10-01311-f002:**
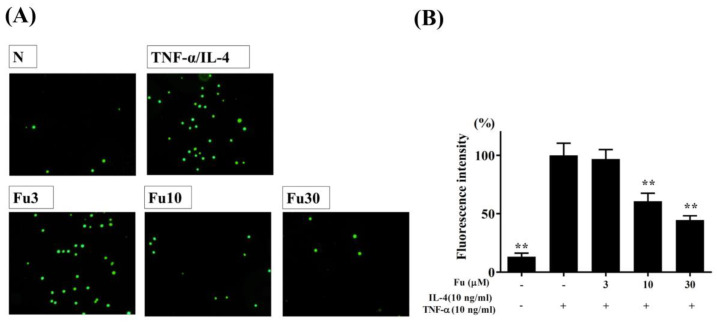
Fucoxanthin (Fu) inhibited THP-1 cell adherence to activated BEAS-2B cells. (**A**) Fluorescence microscopy images of THP-1 cells labeled with calcein-AM and mixed with normal (N) and TNF-α/IL-4-activated BEAS-2B cells in the absence or presence of Fu. (**B**) Fluorescence intensity of monocytic cell adhesion to BEAS-2B cells. Three independent experiments were analyzed. Data are presented as the mean ± SEM. ** *p* < 0.01 compared to BEAS-2B cells stimulated with TNF-α and IL-4.

**Figure 3 cells-10-01311-f003:**
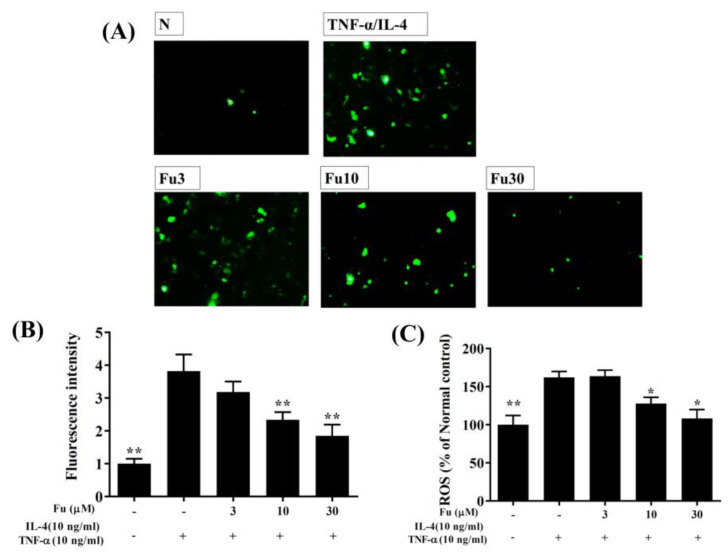
The effects of fucoxanthin (Fu) on ROS production in activated BEAS-2B cells. (**A**) Fluorescence microscopy images of intracellular ROS. (**B**) Fluorescence intensity of intracellular ROS. (**C**) Percentages of ROS detected in TNF-α/IL-4-activated BEAS-2B cells in the absence or presence of Fu compared to untreated cells (N). Three independent experiments were analyzed. Data are presented as mean ± SEM. * *p* < 0.05, ** *p* < 0.01 compared to BEAS-2B cells stimulated with TNF-α and IL-4.

**Figure 4 cells-10-01311-f004:**
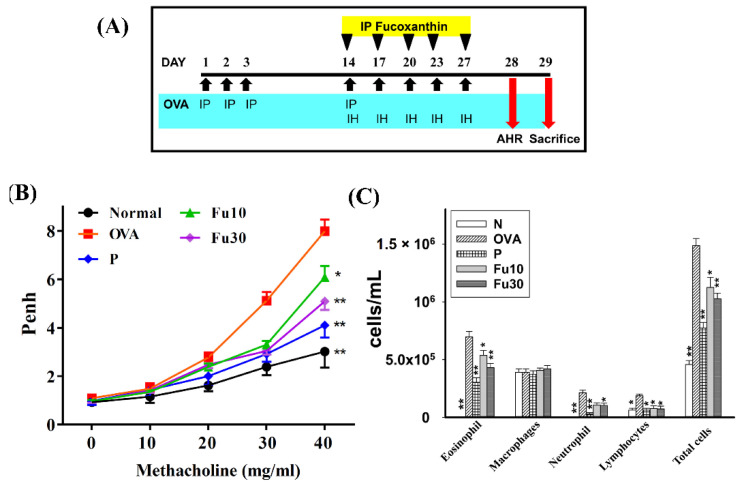
The effect of fucoxanthin (Fu) on AHR and cell counts in the BALF of asthmatic mice. (**A**) On days 1–3 and 14, the mice were sensitized with OVA via intraperitoneal injection (IP) and then challenged with 2% OVA inhalation (IH) on days 14, 17, 20, 23, and 27. One hour before the OVA challenge or methacholine inhalation, the mice were treated with fucoxanthin or DMSO (*n* = 10 mice/group) via intraperitoneal injection. (**B**) The mice inhaled increasing doses of methacholine and AHR, assessed as Penh values. (**C**) The inflammatory cells in the BALF were measured. Three independent experiments were analyzed. Data are presented as mean ± SEM. * *p* < 0.05, ** *p* < 0.01 compared to the OVA control group. 10 mg/kg and 30 mg/kg fucoxanthin were named as Fu10 and Fu30, respectively. 5 mg/kg prednisolone was named as P.

**Figure 5 cells-10-01311-f005:**
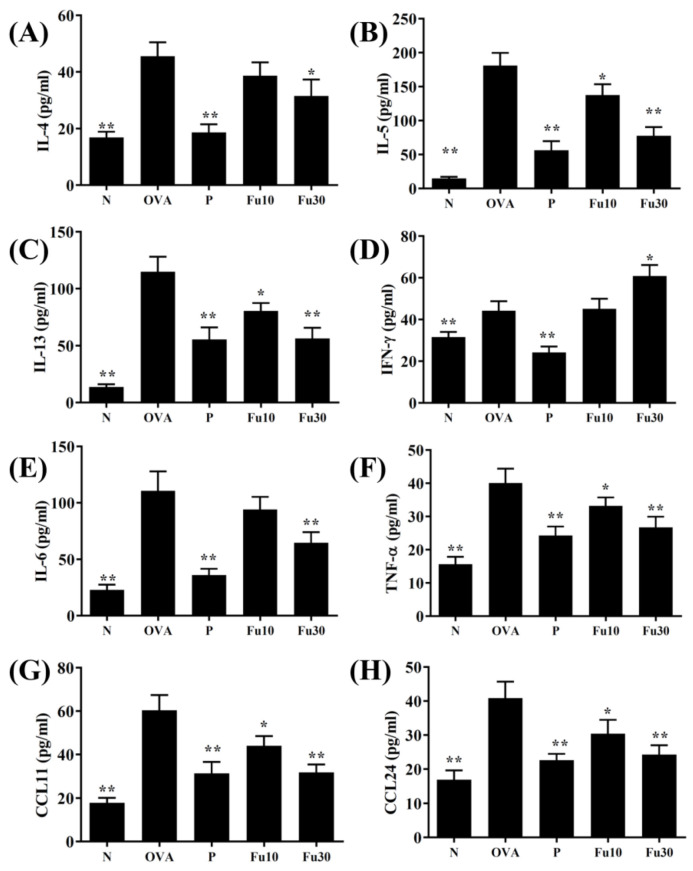
The effects of fucoxanthin (Fu) on the cytokine and chemokine levels in the BALF. (**A**) The concentrations of IL-4, (**B**) IL-5, (**C**) IL-13, (**D**) IFN-γ, (**E**) IL-6, (**F**) TNF-α, (**G**) CCL11, and (**H**) CCL24 were measured by ELISA using BALF from normal (N) and OVA-stimulated (OVA) mice with or without Fu (10 or 30 μM) treatment. Three independent experiments were analyzed. Data are presented as mean ± SEM. * *p* < 0.05, ** *p* < 0.01 compared to the OVA control group. 10 mg/kg and 30 mg/kg fucoxanthin were named as Fu10 and Fu30, respectively. 5 mg/kg prednisolone was named as P.

**Figure 6 cells-10-01311-f006:**
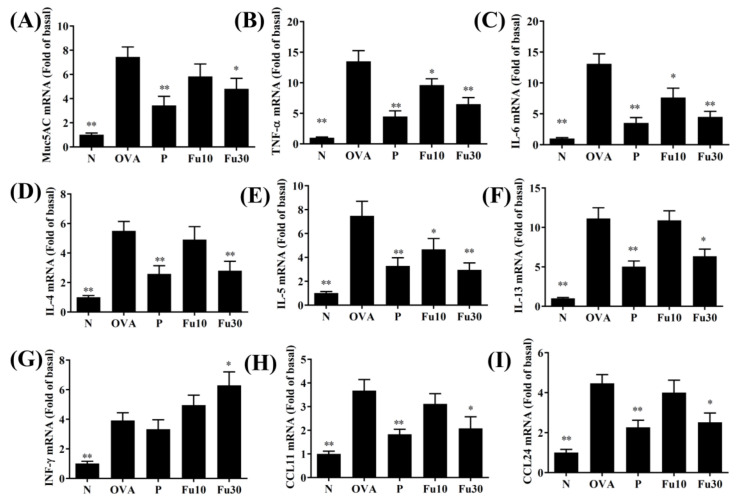
The effects of fucoxanthin (Fu) on cytokine, chemokine, and inflammatory mediator mRNA expression in the lungs. Gene expression levels were determined by the real-time RT-PCR of RNA extracted from the lung tissues of normal (N) and OVA-stimulated (OVA) mice with or without Fu (10 or 30 μM) treatment. (**A**) Muc5AC, (**B**) TNF-α, (**C**) IL-6, (**D**) IL-4, (**E**) IL-5, (**F**) IL-13, (**G**) IFN-γ, (**H**) CCL11, and (**I**) CCL24. Fold changes in expression were measured relative to β-actin (internal control). Three independent experiments were analyzed. Data are presented as mean ± SEM. * *p* < 0.05, ** *p* < 0.01 compared to the OVA control group. 10 mg/kg and 30 mg/kg fucoxanthin were named as Fu10 and Fu30, respectively. 5 mg/kg prednisolone was named as P.

**Figure 7 cells-10-01311-f007:**
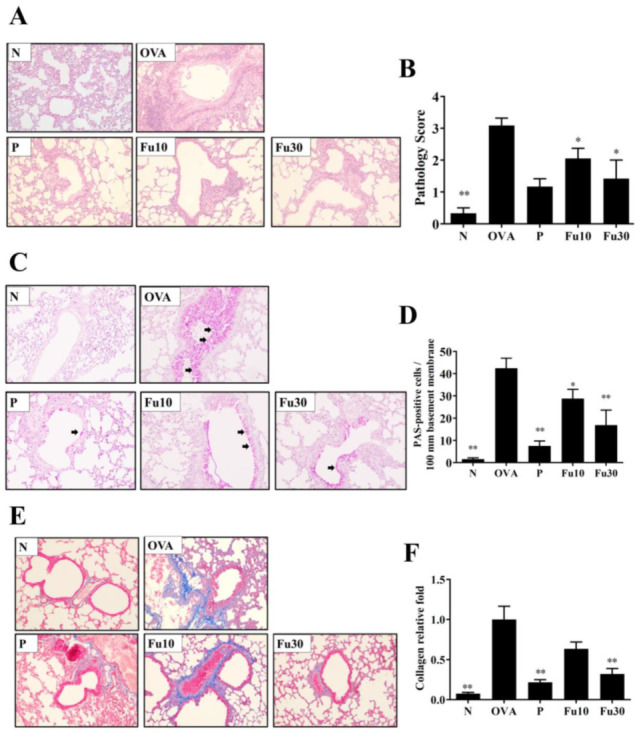
The effects of fucoxanthin (Fu) on asthmatic lung tissue. Histological sections of lung tissues from normal (N) and OVA-stimulated (OVA) mice with or without Fu (10 or 30 μM) treatment. (**A**) Fu reduced eosinophil infiltration. HE stain, 200× magnification. (**B**) Inflammation was scored by a pathological evaluation of inflammatory cell infiltration in lung sections. (**C**) PAS-stained lung sections show goblet cell hyperplasia. Goblet cells are indicated by arrows. 200× magnification. (**D**) Results were expressed as the number of PAS-positive cells per 100 μm of the basement membrane. (**E**) Lung sections were stained with Masson’s trichrome stain to detect collagen expression. 200× magnification. (**F**) Quantitative analysis of collagen in lung sections. Three independent experiments were analyzed. Data are presented as mean ± SEM. * *p* < 0.05, ** *p* < 0.01 compared to the OVA control group. 10 mg/kg and 30 mg/kg fucoxanthin were named as Fu10 and Fu30, respectively. 5 mg/kg prednisolone was named as P.

**Figure 8 cells-10-01311-f008:**
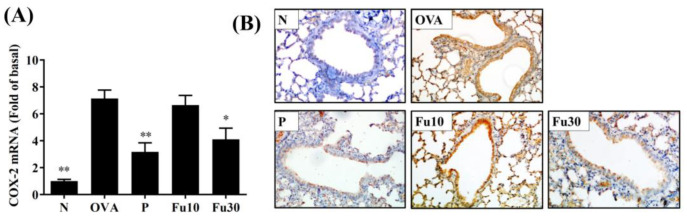
The effects of fucoxanthin (Fu) on COX-2 expression in the lung tissues from OVA-sensitized mice. (**A**) COX-2 gene expression was detected by real-time RT-PCR and fold-changes in expression measured relative to β-actin (internal control). (**B**) COX-2 expression was analyzed by immunohistochemistry staining and labeled as a brown colored drop. Three independent experiments were analyzed. Data are presented as mean ± SEM. * *p* < 0.05, ** *p* < 0.01 compared to the OVA control group. 10 mg/kg and 30 mg/kg fucoxanthin were named as Fu10 and Fu30, respectively. 5 mg/kg prednisolone was named as P.

**Figure 9 cells-10-01311-f009:**
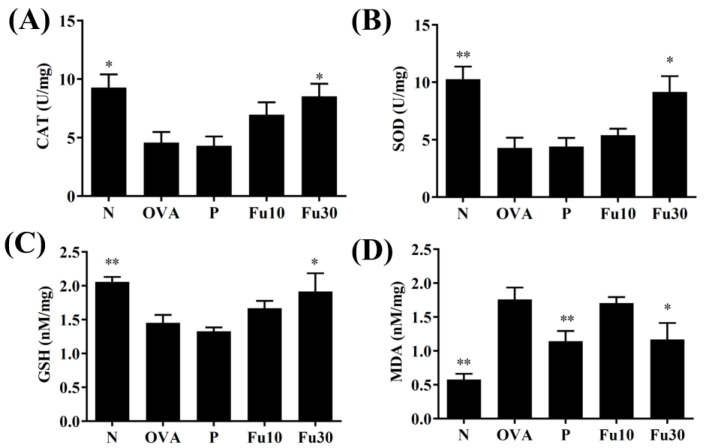
The effect of fucoxanthin (Fu) on oxidative stress factors. (**A**) CAT, (**B**) SOD, (**C**) GSH, and (**D**) MDA activity in the lung tissues from the mice. Three independent experiments were analyzed. Data are presented as mean ± SEM. * *p* < 0.05, ** *p* < 0.01 compared to the OVA control group. 10 mg/kg and 30 mg/kg fucoxanthin were named as Fu10 and Fu30, respectively. 5 mg/kg prednisolone was named as P.

**Figure 10 cells-10-01311-f010:**
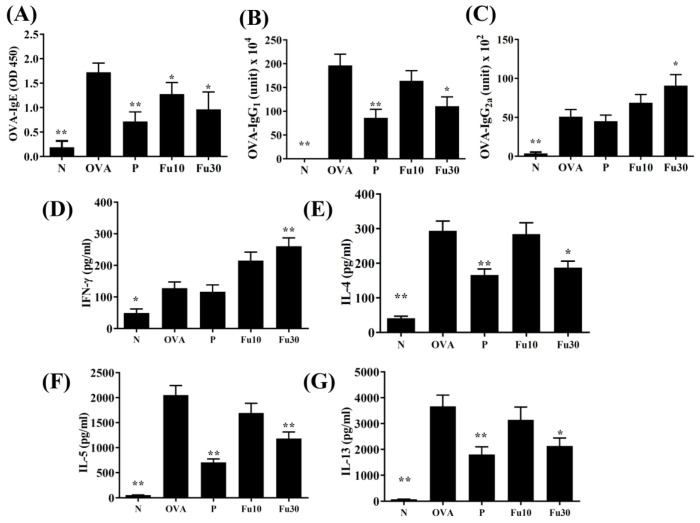
The Fucoxanthin (Fu) effects on OVA-specific antibodies in serum. The serum levels of (**A**) OVA-IgE, (**B**) OVA-IgG_1_, and (**C**) OVA-IgG_2a_ are shown for normal (N) and OVA-stimulated (OVA) mice treated without or with prednisolone (P) or fucoxanthin (Fu10 and Fu30). Fu modulated the levels of (**D**) IFN-γ, (**E**) IL-4, (**F**) IL-5, and (**G**) IL-13 produced by OVA-activated splenocytes. All data are the means ± SEM. * *p* <0.05, ** *p* <0.01 compared to the OVA control group. 10 mg/kg and 30 mg/kg fucoxanthin were named as Fu10 and Fu30, respectively. 5 mg/kg prednisolone was named as P.

## Data Availability

The data presented in this study are available on request from the corresponding author.

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
