# Peer review of "Fucoxanthin Ameliorates Oxidative Stress and Airway Inflammation in Tracheal Epithelial Cells and Asthmatic Mice"

_cells, 2021, doi:10.3390/cells10061311_

Round 1
Reviewer 1 Report
The manuscript entitled ‘Fucoxanthin ameliorates oxidative stress and airway inflammation in tracheal epithelial cells and asthmatic mice’ is an original article. It concerns a potential use of fucoxanthin in asthma treatment. In the experiments mouse model of asthma is used. The authors included the ethical approval number of relevant committee. The manuscript is well written, the design of the study and controls are correct. It contains well described figures. The dosage of fucoxanthin is well established. Altogether, the manuscript provides precious information on the mechanisms of protection and possibilities of application of fucoxanthin, a natural compound, in the treatment of asthma - a civilization disease. I do recommend this article for publication. Some minor improvements are required: Line 36: Please, write the abbreviation for malondialdehyde to be consistent with the way of implementation of other abbreviations. Line 131: Expression “using specific ELISA kits” – please using brackets add the names of kits and manufactures. Line 420: Sentence ‘In the current study, fucoxanthin promoted 420 SOD, CAT, and GSH expression and decreased MDA expression in the lungs of asthmatic mice. Thus, fucoxanthin could regulate and improve lung peroxidation in asthmatic mice’ – please, refer to appropriate Figure(s) at the end of the sentence. Line 155: Please, add in brackets what prednisolone is (just several words). Line 438: Sentence ‘Our experiments showed that the lung tissues, BALF, and spleen cell culture medium for asthmatic mice had significantly reduced IL-13 levels and reduced AHR values after treatment with fucoxanthin’ – refer to appropriate Figure(s) at the end of the sentence. Line 456: Once again, sentence ‘In our asthmatic mouse model, the BALF and lungs express higher IL-5 and eotaxin levels than normal mice’ - refer to appropriate Figure(s) at the end of the sentence. Line 463: ‘Fucoxanthin reduced the expression of ICAM-1 in inflamed tracheal epithelial cells, reducing the adherence of THP-1 cells to tracheal epithelial cells’ - refer to appropriate Figure(s) at the end of the sentence. Line 474: ‘Our experiments used PAS staining to observe the goblet cell hyperplasia in asthmatic mice, and demonstrated that fucoxanthin significantly reduces tracheal goblet cell hyperplasia and decreases Muc5Ac gene expression in the lungs of asthmatic mice’ - refer to appropriate Figure(s) at the end of the sentence. Line 491: ‘In the current study, fucoxanthin significantly inhibited collagen deposition in the lungs of asthmatic mice’ - refer to appropriate Figure(s) at the end of the sentence.Author Response
The manuscript entitled ‘Fucoxanthin ameliorates oxidative stress and airway inflammation in tracheal epithelial cells and asthmatic mice’ is an original article. It concerns a potential use of fucoxanthin in asthma treatment. In the experiments mouse model of asthma is used. The authors included the ethical approval number of relevant committee. The manuscript is well written, the design of the study and controls are correct. It contains well described figures. The dosage of fucoxanthin is well established. Altogether, the manuscript provides precious information on the mechanisms of protection and possibilities of application of fucoxanthin, a natural compound, in the treatment of asthma - a civilization disease. I do recommend this article for publication. Some minor improvements are required:
1.Line 36: Please, write the abbreviation for malondialdehyde to be consistent with the way of implementation of other abbreviations.
Responses:
Thank for reviewer’s suggestion. We have also carefully checked the abbreviations in this manuscript.
2.Line 131: Expression “using specific ELISA kits” – please using brackets add the names of kits and manufactures.
Responses:
We added the names of kits and manufactures in line 201-204.
“Supernatant from BEAS-2B cell culture medium and BALF was used to detected 201 CCL5, CCL11, CCL24, MCP-1, intercellular adhesion molecule 1 (ICAM-1), TNF-α, IL-4, 202 IL-5, IL-6, IL-8, and IL-13 levels using specific ELISA kits (R&D Systems, Minneapolis, 203 MN, USA)”
3.Line 420: Sentence ‘In the current study, fucoxanthin promoted SOD, CAT, and GSH expression and decreased MDA expression in the lungs of asthmatic mice. Thus, fucoxanthin could regulate and improve lung peroxidation in asthmatic mice’ – please, refer to appropriate Figure(s) at the end of the sentence.
Responses:
Thank for reviewer’s suggestion. We modified this sentence as “In the current study, fucoxanthin promoted SOD, CAT, and GSH expression and decreased MDA expression in the lungs of asthmatic mice (Figure 9)”.
4.Line 155: Please, add in brackets what prednisolone is (just several words).
Responses:
We modified this sentence in line 155
5.Line 438: Sentence ‘Our experiments showed that the lung tissues, BALF, and spleen cell culture medium for asthmatic mice had significantly reduced IL-13 levels and reduced AHR values after treatment with fucoxanthin’ – refer to appropriate Figure(s) at the end of the sentence.
Responses:
We modified this sentence as “Our experiments showed that the lung tissues, BALF, and spleen cell culture medium for asthmatic mice had significantly reduced IL-13 levels and reduced AHR values after treatment with fucoxanthin (Figure 4-5, 10)”.
5.Line 456: Once again, sentence ‘In our asthmatic mouse model, the BALF and lungs express higher IL-5 and eotaxin levels than normal mice’ - refer to appropriate Figure(s) at the end of the sentence.
Responses:
We modified this sentence as “In our asthmatic mouse model, the BALF and lungs express higher IL-5 and eotaxin levels than normal mice (Figure 5-6).”
6.Line 463: ‘Fucoxanthin reduced the expression of ICAM-1 in inflamed tracheal epithelial cells, reducing the adherence of THP-1 cells to tracheal epithelial cells’ - refer to appropriate Figure(s) at the end of the sentence.
Responses:
We modified this sentence as “Fucoxanthin reduced the expression of ICAM-1 in inflamed tracheal epithelial cells, reducing the adherence of THP-1 cells to tracheal epithelial cells (Figure 2).”
7.Line 474: ‘Our experiments used PAS staining to observe the goblet cell hyperplasia in asthmatic mice, and demonstrated that fucoxanthin significantly reduces tracheal goblet cell hyperplasia and decreases Muc5Ac gene expression in the lungs of asthmatic mice’ - refer to appropriate Figure(s) at the end of the sentence.
Responses:
We modified this sentence as “ Our experiments used PAS staining to observe the goblet cell hyperplasia in asthmatic mice, and demonstrated that fucoxanthin significantly reduces tracheal goblet cell hy-perplasia and decreases Muc5Ac gene expression in the lungs of asthmatic mice (Figure 6-7).”
8.Line 491: ‘In the current study, fucoxanthin significantly inhibited collagen deposition in the lungs of asthmatic mice’ - refer to appropriate Figure(s) at the end of the sentence.
Responses:
We modified this sentence as “ In the current study, fucoxanthin significantly inhibited collagen deposition in the lungs of asthmatic mice (Figure 7).”
Reviewer 2 Report
To the authors:
- General comments:
The article entitled “Fucoxanthin ameliorates oxidative stress and airway inflammation in tracheal epithelial cells and asthmatic mice” written by the authors: Shun-Ju Wu, Chian-Jiun Liou, Ya-Ling Chen, Shu-Chen Cheng, and Wen-Chung Huang describes the investigation of the effects of fucoxanthin on oxidative and inflammatory responses in inflammatory human tracheal epithelial BEAS-2B cells and attenuated airway hyperresponsiveness, airway inflammation and oxidative stress in asthma mice. I enjoyed the read and I consider this work is very important and interesting for perform further studies about this compound. I have just some minor comments:
Specific comments for revision:
- Lines 184. I suggest the authors to specify in this section the details of the centrifugation of the blood.
- Line 217. I suggest completing the information about the perchloric acid used in protein precipitation with concentration and volume.
- Line 224. I suggest explaining a little more the statistical analysis performed. Authors should include a sentence saying that the normality was asses by using the selected mathematical test to prove the normality. As the mice are 10 per group if the normality was not proved, the ANOVA was wrongly used, as is a parametric test. The correct test to use in the case of no normality is Kruskal-Wallis. In addition, I recommend using minus “p” and italic to the “p value” in all the manuscript and figures. I also recommend specifying the abbreviation SEM in this section.
- Figure 7. I am confused about the Line 342 “Goblet cells are indicated by arrows” I did not see any arrow in the pictures. Please guide the reader better here.
- Figures in General. Please review the figures as not in all of them “P” for prednisolone are described in the figure description. Example in Figure 7.
Author Response
General comments:
The article entitled “Fucoxanthin ameliorates oxidative stress and airway inflammation in tracheal epithelial cells and asthmatic mice” written by the authors: Shun-Ju Wu, Chian-Jiun Liou, Ya-Ling Chen, Shu-Chen Cheng, and Wen-Chung Huang describes the investigation of the effects of fucoxanthin on oxidative and inflammatory responses in inflammatory human tracheal epithelial BEAS-2B cells and attenuated airway hyperresponsiveness, airway inflammation and oxidative stress in asthma mice. I enjoyed the read and I consider this work is very important and interesting for perform further studies about this compound. I have just some minor comments:
Specific comments for revision:
Lines 184. I suggest the authors to specify in this section the details of the centrifugation of the blood.
Responses:
Thank for reviewer’s suggestion. We modified those sentences as” Blood centrifuged at 6,000 rpm for 5 min; then, the serum was collected and stored at −80°C. Serum would detect OVA-specific antibody expression by ELISA as described previously”.
Line 217. I suggest completing the information about the perchloric acid used in protein precipitation with concentration and volume.
Responses:
Thank for reviewer’s suggestion. We modified those sentences as”
Briefly, lung tissues were homogenized using an homogenizer (FastPrep-24, MP Biomedicals, Santa Ana, CA, USA) and treated with 150 μl 2N perchloric acid for protein precipitation.”
Line 224. I suggest explaining a little more the statistical analysis performed. Authors should include a sentence saying that the normality was asses by using the selected mathematical test to prove the normality. As the mice are 10 per group if the normality was not proved, the ANOVA was wrongly used, as is a parametric test. The correct test to use in the case of no normality is Kruskal-Wallis. In addition, I recommend using minus “p” and italic to the “p value” in all the manuscript and figures. I also recommend specifying the abbreviation SEM in this section.
Responses:
Thank for reviewer’s suggestion. We modified the statistical methods as ANOVA followed by a Kruskal-Wallis test. We also modified reviewer’s suggestion, including p value and standard error of the mean (SEM).
Figure 7. I am confused about the Line 342 “Goblet cells are indicated by arrows” I did not see any arrow in the pictures. Please guide the reader better here.
Responses:
Thank for reviewer’s suggestion. We modified the Figure 7C to add arrows.
Figures in General. Please review the figures as not in all of them “P” for prednisolone are described in the figure description. Example in Figure 7.
Responses:
Thank for reviewer’s suggestion. We add those descriptions as “10 mg/kg, 30 mg/kg fucoxanthin were named as Fu10 and Fu30, respectively. 5 mg/kg prednisolone was named as P.” in Figure legend 4-10.